# Alterations of Exercise-Induced Carbohydrate and Fat Oxidation by Anthocyanin-Rich New Zealand Blackcurrant Are Associated with the Pre-Intervention Metabolic Function: A Secondary Analysis of Randomized Crossover Trials

**DOI:** 10.3390/nu17060997

**Published:** 2025-03-12

**Authors:** Mark E. T. Willems, Matthew D. Cook

**Affiliations:** 1Institute of Applied Sciences, University of Chichester, College Lane, Chichester PO19 6PE, UK; 2School of Sport & Exercise Science, University of Worcester, Henwick Grove, Worcester WR2 6AJ, UK; matthew.cook@worc.ac.uk

**Keywords:** respiratory exchange ratio, fat oxidation, carbohydrate oxidation, polyphenols, *Ribes nigrum* L.

## Abstract

**Background/Objectives**: Our studies have provided evidence for the alteration of exercise-induced metabolic responses by the intake of anthocyanin-rich New Zealand blackcurrant (NZBC) extract. In this secondary analysis of 10 studies, we examined the relationship between the pre-intervention exercise-induced respiratory exchange ratio and the blackcurrant-induced respiratory exchange ratio and substrate utilisation during exercise. **Methods**: Metabolic data of seven cohort and three case studies with females (n = 46) and males (n = 71), from recreationally active to ultra-endurance trained individuals that were dosed with different intake durations (acute to two-week intake) and dosages (105 to 420 mg of anthocyanins) of NZBC extract for walking-, running-, and cycling-induced effects, were included in the secondary analysis. **Results**: There was a strong positive correlation between the pre-intervention and blackcurrant-induced respiratory exchange ratio for females (Pearson r: 0.7972, *p* < 0.0001) and males (Pearson r: 0.8674, *p* < 0.0001). A moderate positive correlation was obtained for the relationship between the pre-intervention respiratory exchange ratio and changes in fat oxidation for females (Pearson r: 0.5311, *p* = 0.0001) and males (Pearson r: 0.3136, *p* = 0.002). In addition, a moderate negative correlation was obtained for the relationship between the pre-intervention respiratory exchange ratio and changes in carbohydrate oxidation for females (Pearson r: −0.3017, *p* = 0.0393) and males (Pearson r: −0.3327, *p* < 0.001). There were no differences between females and males in the changes of the exercise-induced metabolic responses to the intake of New Zealand blackcurrant extract. **Conclusions**: Our secondary analysis of the data in studies on the effects of New Zealand blackcurrant extract suggests that the metabolic response of individuals to the intake of New Zealand blackcurrant extract depends partly on the pre-intervention respiratory exchange ratio, with the majority of individuals showing enhanced exercise-induced fat oxidation and lower exercise-induced carbohydrate oxidation. However, a divergent metabolic response seems possible such that individuals with a very low intrinsic respiratory exchange ratio may more likely experience lower fat oxidation and higher carbohydrate oxidation with the intake of New Zealand blackcurrant. Individuals with a high intrinsic respiratory exchange will more likely experience higher fat oxidation and lower carbohydrate oxidation with the intake of New Zealand blackcurrant. Future work is required to examine the factors and mechanisms for the individual variation of the response of exercise-induced substrate utilisation relative to the intake of anthocyanin-rich New Zealand blackcurrant extracts.

## 1. Introduction

Metabolic energy sources that fuel the main ATP-requiring processes in skeletal muscles during physical exercise, i.e., cross-bridge formation [1], cross-bridge relaxation [2], and sarcoplasmic reticulum calcium pumping activity [3], are associated with the intensity and duration of the exercise (for a review, see [4]). For example, for low- and moderate-intensity exercise, the required ATP resynthesis is primarily met by both the oxidation of exogenous and endogenous carbohydrates and adipocyte-located lipids, with only a small contribution of amino acid oxidation [5,6]. The balance between the oxidation of carbohydrates and lipids during exercise can be quantified with the measurements of oxygen uptake and carbon dioxide production using indirect calorimetry and expressed as the unitless respiratory exchange ratio [7]. The respiratory exchange ratio is the ratio of the volume of carbon dioxide divided by the volume of oxygen. A respiratory exchange ratio of 0.707 indicates 100% lipid oxidation and a value of 1.00 indicates 100% carbohydrate oxidation [8]. As such, the respiratory exchange ratio at rest and during low-intensity to moderate-intensity exercise is an indicator of whole-body metabolic function.

In endurance-trained male (n = 45) and female (n = 16) cyclists, the respiratory exchange ratio (and thus whole-body metabolic function) showed substantial variation at rest and during selected short-duration (i.e., 10-min) exercise intensities (i.e., 25%, 50%, and 75% of the peak cycling power output) [9]. For example, at 25% of peak cycling power output, the respiratory exchange ratio ranged from ~0.78 to ~0.94 [9]. It was suggested that differences in training and dietary intake accounted for the variability in the respiratory exchange ratio [9]. In addition, Rothschild et al. [10], in an analysis of 434 studies on the respiratory exchange during submaximal cycling exercise, concluded that the intensity and duration of the exercise, sex, age, training status and pre-exercise muscle glycogen could explain only 60% of the variation in the exercise-induced respiratory exchange ratio.

The alteration of the exercise-induced respiratory exchange ratio and thus the whole-body metabolic function via an experimental intervention (e.g., physical training or nutrition) is primarily an indication of the adaptation in the metabolic pathways for the oxidation of carbohydrates and lipids. For example, a lower exercise-induced respiratory exchange ratio was observed relative to physical training with moderate-intensity continuous exercise [11] and high-intensity interval training (for a review, see [12]), indicating a higher reliance on fat as an energy source. A similar effect is observed with the acute and chronic intake of high-fat diets [13,14]. The effects on the exercise-induced respiratory exchange ratio has also been examined with respect to the acute and chronic intake of dietary supplements. For example, caffeine intake has been reported in some studies to increase exercise-induced fat oxidation (for a systematic review, see [15]). In active winter triathletes (n = 80, 36 females), the 7-day intake (3 × 70 mL·day^−1^ with 6.5 mmol NO_3_^−^ in 70 mL) of beetroot juice lowered the running-induced respiratory exchange ratio at 80%$\dot{V}$O_2peak_ [16]. Since 2009, studies have examined the effects of the intake of New Zealand blackcurrant on responses during exercise and exercise recovery (e.g., [17]). Blackcurrant contains the polyphenol flavonoid anthocyanin, i.e., primarily delphinidin-3-rutinoside, delphinidin-3-glucoside, cyandin-3-rutinoside, and cyanidin-3-glucoside [18]. Anthocyanins and anthocyanin-induced metabolites can provide anti-oxidant and vasodilatory responses [17,19].

We have reported in several cohort studies on the effects of the intake of New Zealand blackcurrant extracts in altering exercise-induced fat metabolism and the exercise-induced respiratory exchange ratio (i.e., [20,21,22,23,24,25,26]) and also in case studies with ultra-endurance athletes (i.e., [27,28,29]. In our cohort studies, the alteration in the exercise-induced substrate utilisation and respiratory exchange ratio with the intake of New Zealand blackcurrant extract was reported as group effects. It was not examined, in the studies with New Zealand blackcurrant extracts, whether the individual responses of an alteration in a metabolic parameter were associated with the pre-intervention respiratory exchange ratio and whether there is a difference in responses between females and males. The secondary analysis of these studies provides a sample size of 46 females and 71 males that would allow a valid correlation analysis of the exercise-induced metabolic responses to the intake of New Zealand blackcurrant extracts.

Therefore, the aim of the present study was to conduct a secondary analysis of our cohort and case studies on the effects of New Zealand blackcurrant extracts on exercise-induced substrate utilisation to examine the following: (1) the relationship between the pre-intervention (i.e., placebo or control condition) respiratory exchange ratio and the post-intervention respiratory exchange ratio; (2) the relationship between the pre-intervention (i.e., placebo or control condition) respiratory exchange ratio and changes in carbohydrate and fat oxidation; and (3) whether females and males differ in the relationship between the pre-intervention respiratory exchange ratio and the metabolic responses via the intake of New Zealand blackcurrant extracts. The outcome of the secondary analysis of our studies will contribute to the understanding of individual responses to an anthocyanin-rich supplement and may have applications for personalized advice on the intake of New Zealand blackcurrant extracts. The primary focus of the present secondary analysis is on the relationships between metabolic parameters and not an examination of the differences in absolute changes in the metabolic parameters in the cohort and case studies. It was hypothesized that the exercise-induced metabolic response to the intake of anthocyanin-rich New Zealand extracts is associated with the intrinsic exercise-induced respiratory exchange ratio.

## 2. Materials and Methods

### 2.1. Study Design

The present study is a secondary analysis of the respiratory exchange ratio and substrate oxidation individual data from our published cohort and case studies on the effects of the intake of New Zealand blackcurrant extracts on exercise-induced substrate utilisation (i.e., cohort studies [20,21,22,23,24,25,26] and case studies [27,28,29]). All studies had ethical approval from research ethics committees, with all participants providing written informed consent. The individual data of 46 females (46 observations) and 71 males (141 observations) were included in the secondary analysis. Note that the 141 observations were due to three male cohort studies, with one having 7- and 14-day intake durations [23], one with three different intensities [20], and one with two doses [21] used for analyses. For the cohort and case studies, details on the participants, experimental design, dosing strategies with New Zealand blackcurrant extracts, exercise tasks, and observations for the changes in the respiratory exchange ratio are provided in Table 1. The respiratory exchange ratio is calculated as carbon dioxide production (i.e., $\dot{V}$CO_2_) divided by oxygen consumption (i.e., $\dot{V}$O_2_).

Online respiratory gas measurement systems (Moxus metabolic system, AEI Technologies, Pittsburg, PA, USA) were used by Strauss et al. [24] and Cook et al. [22] (Cortex Metalyzer 3B, Biophysik GmbH, Leipzig, Germany). All other studies used Douglas bags for expired air collection with gas analysers and dry gas meters.

In Table 2, we provide the equations that were used in the cohort and case studies to quantify substrate oxidation from the measurement of oxygen consumption and carbon dioxide production.

In each of our cohort studies used for secondary analysis, the relationship between the respiratory exchange ratio and changes in substrate utilisation was not examined due to the sample sizes being considered too small (in all cohort studies, less than 17 participants) to allow valid correlation analyses. In the present study, the amount of data on females (n = 46, i.e., 46 observations) and males (n = 71, i.e., 141 observations) from the published studies [20,21,22,23,24,25,26,27,28,29] will also allow an examination of sex differences in the exercise-induced metabolic response via the intake of New Zealand blackcurrant extracts. We refer to the published studies for additional details on the methodology.

### 2.2. Statistical Analysis

For all female and male participants in the cohort and case studies, Pearson correlation coefficients were calculated and analysed (Prism 5, Graphpad Software, Boston, MA, USA) for the significance of the relationships between metabolic parameters, e.g., between the pre-intervention exercise-induced respiratory exchange ratio and the change in exercise-induced fat oxidation with the intake of New Zealand blackcurrant extracts. Z-scores and probability values were calculated for the analysis of Pearson correlation coefficients. Linear regression analysis was used for slope differences between females and males. Significance was accepted at *p*-values ≤ 0.05.

## 3. Results

### 3.1. Exercise-Induced Respiratory Exchange Ratio Relationships for Females and Males

Figure 1 provides the relationship between the pre-intervention exercise-induced respiratory exchange ratio and the exercise-induced respiratory exchange ratio after New Zealand blackcurrant extract intake for females (a) and males (b). For both females and males, there were significant positive correlations between the exercise-induced respiratory exchange ratios (Pearson r, females: 0.7972 (95%CI [0.6614, 0.8824], *p* < 0.0001; males: 0.8674 (95%CI [0.8196, 0.9032], *p* < 0.0001). There was no difference between the correlations (z-score: −1.338, *p* = 0.09) and the slopes (*p* = 0.608), indicating no sex differences. The location of most of the datapoints below the line of identity in both females and males indicates lower respiratory exchange ratio values with the intake of New Zealand blackcurrant extracts, as reported in the cohort and case studies.

### 3.2. Exercise-Induced Respiratory Exchange Ratio Relationships in the Placebo/Control Condition and Changes in Exercise-Induced Fat Oxidation with the Intake of New Zealand Blackcurrants in Females and Males

Figure 2 provides the relationship between the pre-intervention exercise-induced respiratory exchange ratio and the change in exercise-induced fat oxidation with New Zealand blackcurrant extracts for females (a) and males (b). The correlation equation for changes in exercise-induced fat oxidation crossed the *x*-axis at ~0.80. Less than 10% of the females and 5% of the males had a pre-intervention respiratory exchange ratio lower than 0.80. A respiratory exchange of 0.80 and lower indicates a contribution of fat oxidation of ~70% or more. For both females and males, there were significant positive correlations between the exercise-induced respiratory exchange ratio in the pre-intervention and changes in exercise-induced fat oxidation in the New Zealand blackcurrant extract conditions (Pearson r, females: 0.5311 (95%CI [0.2878, 0.7100], *p* = 0.0001; males: 0.3136 (95%CI [0.1563, 0.4553], *p* = 0.0002). There were no differences between the correlations (z-score: 1.543, *p* = 0.06) and the slopes (*p* = 0.720).

### 3.3. Exercise-Induced Respiratory Exchange Ratio Relationships in the Placebo/Control Condition and Changes in Exercise-Induced Carbohydrate Oxidation with the Intake of New Zealand Blackcurrant in Females and Males

Figure 3 provides the relationship between the pre-intervention exercise-induced respiratory exchange ratio and the change in exercise-induced carbohydrate oxidation with New Zealand blackcurrant extracts for females (a) and males (b). The correlation equation for changes in exercise-induced carbohydrate oxidation also crossed the *x*-axis at ~0.80. A respiratory exchange of 0.80 and lower indicates a carbohydrate oxidation contribution of ~30% or less. For both females and males, there were significant negative correlations between the pre-intervention exercise-induced respiratory exchange ratio and changes in carbohydrate oxidation in the New Zealand blackcurrant extract conditions (Pearson r, females: −0.3017 (95%CI [−0.5419, −0.0158], *p* = 0.0393; males: −0.3327 (95%CI [−0.4633, −0.1662], *p* < 0.001). There was no difference between the correlations (z-score: 0.199, *p* = 0.421) and the slopes (*p* = 0.662), indicating no sex difference.

## 4. Discussion

The secondary analysis of the observations in our published cohort and case studies on the exercise-induced metabolic responses to the intake of New Zealand blackcurrant provides evidence that the responses are related to the pre-intervention respiratory exchange ratio. The pre-intervention respiratory exchange ratio values were normally distributed (D’Agostino and Pearson normality test) for females and males with respect to the means, SDs, and range values (female: 0.87 ± 0.05, range: 0.73–1.00; males: 0.88 ± 0.05, range: 0.77–1.00). The present observations with the variation of the pre-intervention exercise-induced respiratory exchange ratio confirm what has been reported in other studies (e.g., 45 male and 16 female endurance-trained cyclists, range: 0.82–0.98 during cycling exercise at 50% of peak power output [9], ~3304 RER observations from 434 studies [10]). With the data available in the present secondary analysis, there was no evidence of a difference between females and males for the changes in metabolic responses relative to the intake of anthocyanin-rich New Zealand blackcurrant. As far as we know, this observation confirms the findings of the study by Cook et al. [22], the only study in which females and males were examined for their responses to the intake of New Zealand blackcurrant. The findings of the present study suggest that existing differences in metabolic function during moderate-intensity exercise between females and males (e.g., for a review, see [30]) do not result in a gender-specific metabolic response to the intake of New Zealand blackcurrant extracts.

However, the positive and negative correlations for exercise-induced fat oxidation and exercise-induced carbohydrate oxidation as a function of the pre-intervention respiratory exchange ratio may suggest that individuals (female and males) with a very low pre-intervention exercise-induced respiratory exchange ratio may not respond to the intake of New Zealand blackcurrant or even respond with enhanced carbohydrate oxidation and lower fat oxidation. Future work is needed to address the specific mechanisms for the absence and the interindividual metabolic responses to the intake of anthocyanins-rich New Zealand blackcurrant or other anthocyanin-rich supplements. When individuals respond with enhanced fat oxidation, it is possible due to enhanced lipolysis [24] and enhanced blood flow [31], allowing the increased delivery of free fatty acids to contracting skeletal muscles.

The absence of the effects of New Zealand blackcurrant in some individuals in our published cohort and case studies may be due to the dose (primarily 105 and 210 mg of anthocyanins) and intake duration (all studies less than 15 days). In our cohort studies on the effects of New Zealand blackcurrant extracts, i.e., [20,21,22,23,24,25,26], the individual metabolic responses to the intake of New Zealand blackcurrant extracts were not reported. None of our studies have examined the repeatability of the individual metabolic responses using the same dose and intake duration with respect to the New Zealand blackcurrant extract. In addition, human studies on the effects of wild blueberries [32], elderberry juice [33], epigallocatechin-3-gallate and resveratrol [34,35], *Helichrysum italicum* ssp. *italicum* [36], caffeine and oolong tea [37], mixed flavonoids and caffeine [38], blackberry [39], chlorogenic acids [40], licorice flavonoid oil [41], decaffeinated green tea extract [42], green tea extract [43,44,45], and matcha green tea [46,47] have reported enhanced fat oxidation. However, the individual responses in those studies in relation to pre-intervention metabolic functions were not reported. Also, in sport and exercise nutrition studies, studies on the repeatability of metabolic responses via a nutritional ergogenic intervention are limited (e.g., [48]). As far as we know, only a recent study by Perkins et al. [49] reported the repeatability of lactate responses for high-intensity intermittent treadmill running via the intake of New Zealand blackcurrant extracts.

The exercise-induced respiratory exchange ratio is affected by many factors (for a review, see Rothschild et al. [10]). Among the easily modifiable factors are dietary factors, i.e., habitual carbohydrate and fat intake, pre-exercise dietary intake, carbohydrate intake during exercise, and the type of carbohydrate that is being consumed, but there is no mention of the potential effect of supplements [10]. The present observations seem to suggest that there may be a cross-over for the effect on metabolic responses via the intake of New Zealand blackcurrant extracts at a low respiratory exchange ratio. Intuitively, for individuals who already have a high internal ability for fat oxidation, the mechanisms and potential changes for the alteration of fat oxidation enhancement may be less sensitive to the effects of anthocyanins or anthocyanin-induced metabolites. The respiratory exchange ratio is related to the oxidative capacity of skeletal muscles, and when this oxidative capacity favours fat as an energy source, i.e., at a very low respiratory exchange ratio, the muscle is likely to be highly adapted for fat oxidation. In general, physical training can enhance exercise-induced fat oxidation related to proteins that control intramuscular lipid storage, the mobilization of free fatty acids, and fat oxidation [50,51]. However, physical training studies will normally examine changes in proteins that happen over a long period of time, whereas studies with nutritional interventions that showed enhanced fat oxidation sometimes use relatively short intake durations (e.g., 2 weeks [23]), pointing to the different mechanisms for enhancing fat oxidation via nutritional and physical training interventions. Future cohort studies on exercise-induced metabolic responses via the intake of New Zealand blackcurrant extracts and other anthocyanin-rich supplements are encouraged to report on the individual metabolic responses in relation to the pre-intervention exercise-induced respiratory exchange ratio.

However, a few limitations need to be noted. First, none of the cohort and case studies specifically recruited individuals with a particular respiratory exchange ratio. Second, the data collection and analysis methods for obtaining metabolic data were different between the studies. Third, none of the studies implemented strict dietary control, which may have affected the absorption, distribution, metabolism, and excretion of anthocyanins and anthocyanin-induced metabolites. Fourth, age and training status were not considered. As the precise mechanisms and factors that contribute to the alteration of exercise-induced substrate utilisation via the intake of New Zealand blackcurrant are unknown, our secondary analysis did not have statistical adjustments for confounding variables. In addition, the 10 studies that were included in the analysis all had a cross-over design and randomization, minimizing the effect of any (but unknown) confounding variables. Therefore, our secondary analysis indicates that the individual exercise-induced metabolic response to the intake of New Zealand blackcurrant extract is related to the intrinsic respiratory exchange ratio.

## 5. Conclusions

Individual responses to the intake of a dietary supplement in order to affect exercise responses always show variation. In the case of exercise-induced metabolic responses to the intake of New Zealand blackcurrant extracts, a divergent response seems to be possible, such that the individual could be considered a responder with enhanced fat oxidation or enhanced carbohydrate oxidation depending on the pre-intervention respiratory exchange ratio. Individuals with a very low respiratory exchange ratio may not respond to the intake of New Zealand blackcurrant extracts or respond with enhanced carbohydrate oxidation and lower fat oxidation. Our observations could have implications for personalized supplementation strategies with anthocyanin-rich supplements.

## Figures and Tables

**Figure 1 nutrients-17-00997-f001:**
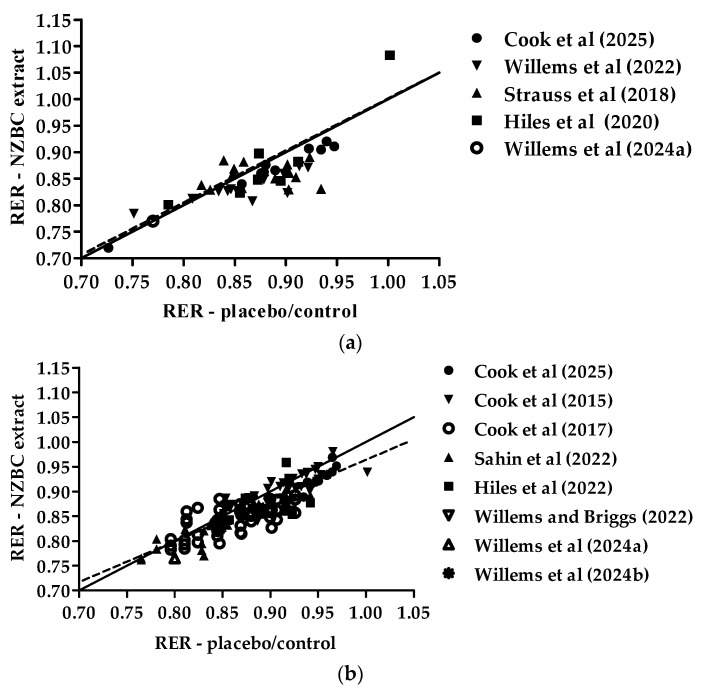
Relationship between the exercise-induced respiratory exchange ratio (RER) in the placebo/control and New Zealand blackcurrant extract conditions for females (**a**) and males (**b**). The dotted lines indicate linear positive correlations. The solid lines are the lines of identity. Datapoints for females (**a**): ●, from Cook et al. [22]; ▼, Willems et al. [26]; ▲, Strauss et al. [24]; ■, Hiles et al. [25]; ◯, Willems et al. [28]. Datapoints males (**b**): ●, Cook et al. [22]; ▼, Cook et al. [20]; ◯, Cook et al. [21]; ▲, Şahin et al. [23]; ■, Hiles et al. [25]; ▽, Willems and Briggs [27]; △, Willems et al. [28]; ✺, Willems et al. [29].

**Figure 2 nutrients-17-00997-f002:**
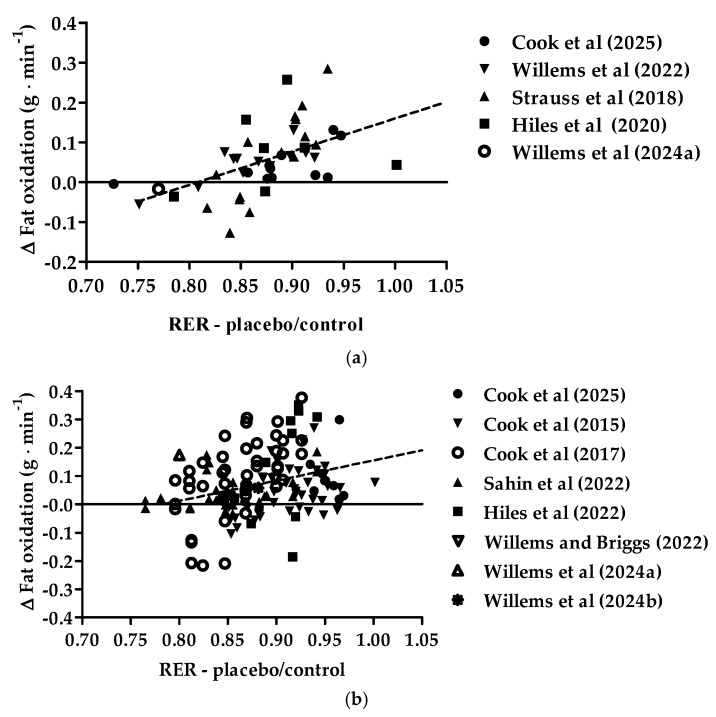
Relationship between the exercise-induced respiratory exchange ratio (RER) in the placebo/control and changes (∆) in exercise-induced fat oxidation with the intake of New Zealand blackcurrant extracts for females (**a**) and males (**b**). The dotted lines indicate the linear positive correlations. Datapoints for females (**a**): ●, from Cook et al. [22]; ▼, Willems et al. [26]; ▲, Strauss et al. [24]; ■, Hiles et al. [25]; ◯, Willems et al. [28]. Datapoints males (**b**): ●, Cook et al. [22]; ▼, Cook et al. [20]; ◯, Cook et al. [21]; ▲, Şahin et al. [23]; ■, Hiles et al. [25]; ▽, Willems and Briggs [27]; △, Willems et al. [28]; ✺, Willems et al. [29].

**Figure 3 nutrients-17-00997-f003:**
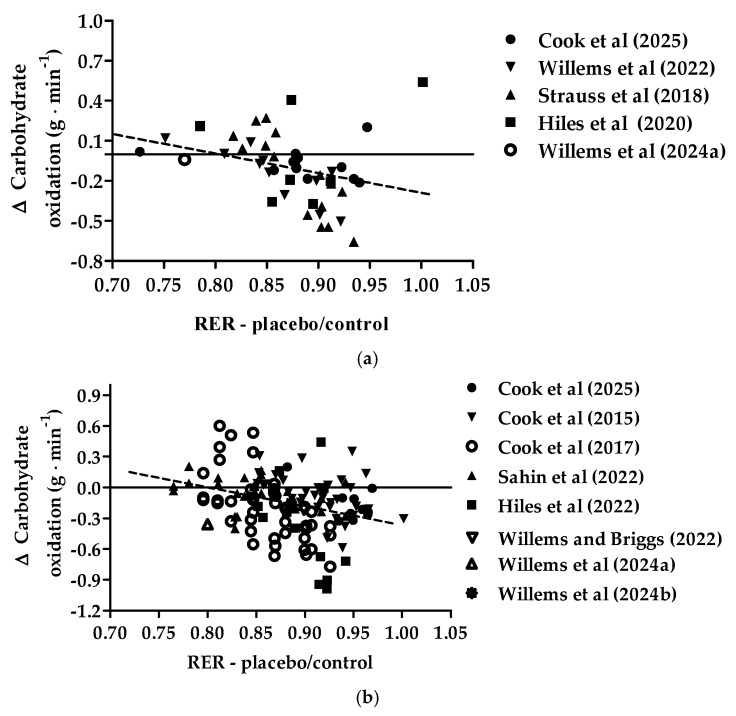
Relationship between the exercise-induced respiratory exchange ratio (RER) in the placebo/control and changes (∆) in exercise-induced carbohydrate oxidation with the intake of New Zealand blackcurrant extracts for females (**a**) and males (**b**). The dotted lines indicate linear negative correlations. Datapoints for females (**a**): ●, from Cook et al. [22]; ▼, Willems et al. [26]; ▲, Strauss et al. [24]; ■, Hiles et al. [25]; ◯, Willems et al. [28]. Datapoints males (**b**): ●, Cook et al. [22]; ▼, Cook et al. [20]; ◯, Cook et al. [21]; ▲, Şahin et al. [23]; ■, Hiles et al. [25]; ▽, Willems and Briggs [27]; △, Willems et al. [28]; ✺, Willems et al. [29].

**Table 1 nutrients-17-00997-t001:** Summary of the methods and observations of the respiratory exchange ratio (RER) of the cohort and case studies examining the effects of the intake of anthocyanin-rich New Zealand blackcurrant (NZBC) extracts on exercise-induced substrate utilisation. The studies were used for the secondary analysis. The cohort and case information in Table 1 were obtained from the authors. MET, metabolic equivalent. ↓ indicates a decrease.

Source	Participant(s)	Design	Dosing Strategy	Exercise Task	RER Outcome
Cook et al. [20]	14 males, cycling 8–10 h per week, age: 38 ± 13 years	double blind, placebo-controlled, randomized, cross-over	105 mg NZBC anthocyanins in extract form for 7 days	$3\;\times \;10\;\min\;\mathrm{cycling}\;\mathrm{at}\;45\%,\;55\%\;\mathrm{and}\;65\%\;\dot{V}$O_2_max	45%: ↓0.01 (*p* = 0.006) 55%: ↓0.02 (*p* = 0.102) 65%: ↓0.01 (*p* = 0.043)
Cook et al. [21]	15 males, cycling 6–10 h per week, age: 38 ± 15 years	randomized, counterbalanced, latin-square	control (no dose), 105, 210, and 315 mg NZBC anthocyanins in extract form for 7 days	$120\;\min\;\mathrm{cycling}\;\mathrm{at}\;65\%\;\dot{V}$O_2_max	210 mg: ↓0.03 (*p* < 0.05) 315 mg: ↓0.02 (*p* < 0.05)
Cook et al. [22]	recreationally active, 11 females, age 27 ± 7 years; 11 males, age: 32 ± 9 years	double blind, placebo-controlled, randomized, cross-over	210 mg NZBC anthocyanins in extract form for 7 days	60 min of treadmill walking at 50% $\dot{V}$O_2_max	males: ↓0.02 (*p* < 0.05) females: ↓0.02 (*p* < 0.05)
Willems et al. [26]	recreationally active, 12 females, age 21 ± 2 years	double blind, placebo-controlled, randomized, cross-over	210 mg NZBC anthocyanins in extract form for 7 days	30 min of treadmill walking at moderate intensity (4.7 ± 0.4 MET)	↓0.03 (*p* = 0.009)
Strauss et al. [24]	endurance trained, 16 females, age: 28 ± 8 years	double blind, placebo-controlled, randomized, cross-over	210 mg NZBC anthocyanins in extract form for 7 days	$120\;\min\;\mathrm{cycling}\;\mathrm{at}\;65\%\;\dot{V}$O_2_max	↓0.02 (*p* = 0.063)
Şahin et al. [23]	16 males, age 24 ± 6 years	randomized, cross-over	control (no dose) and 210 mg NZBC anthocyanins in extract form for 7 and 14 days	30 min of treadmill walking at moderate-intensity (n = 3: 4-MET, n = 13: 5-MET)	7 days: ↓0.009 (*p* = 0.122) 14 days: ↓0.016 (*p* = 0.004)
Hiles et al. [25]	recreationally active; 6 females, 12 males: age 27 ± 6 years	double blind, placebo-controlled, randomized, cross-over	210 mg NZBC anthocyanins in extract form for 7 days	60 min of fasted running at 65% $\dot{V}$O_2_max (34 °C and 40% relative humidity)	↓0.02 (*p* = 0.04)
Willems and Briggs [27]	male amateur ultra-endurance runner, age: 40 years	allocated, cross-over	210 mg NZBC anthocyanins in extract form for 7 days	$120\;\min\;\mathrm{treadmill}\;\mathrm{running}\;\mathrm{at}\;58\%\;\dot{V}$O_2_max (26 °C and relative humidity of 70%)	↓0.02 (*p* < 0.01)
Willems et al. [28]	female (age: 23 years) and male (age: 38 years) amateur Marathon des Sables athletes	randomized, cross-over	control (no dose) and 210 mg NZBC anthocyanins in extract form for 7 days	$60\;\min\;\mathrm{treadmill}\;\mathrm{running}\;\mathrm{at}\;50\%\;\dot{V}$O_2_max (34 °C and relative humidity of 30%)	male: ↓0.03 (*p* = 0.04) female: no change
Willems et al. [29]	male amateur Ironman athlete	single blind, placebo-controlled, randomized, cross-over	acute intake of 420 mg NZBC anthocyanins in extract form	240 min of indoor cycling at 165 Watts	↓0.02 (*p* = 0.042)

**Table 2 nutrients-17-00997-t002:** Equations used in cohort and case studies on the effects of New Zealand blackcurrant extracts on exercise-induced fat and carbohydrate oxidation. $\dot{V}$O_2_, oxygen consumption; $\dot{V}$CO_2_, carbon dioxide production.

Source	Fat Oxidation	Carbohydrate Oxidation
Cook et al. [20]	1.695 × $\dot{V}$CO_2_ − 1.701 × $\dot{V}$CO_2_	Low intensity: 4.344 × $\dot{V}$CO_2_ − 3.061 × $\dot{V}$O_2_
		Moderate intensity: 4.210 × $\dot{V}$CO_2_ − 2.962 × $\dot{V}$O_2_
Cook et al. [21], Willems et al. [26], Strauss et al. [24], Şahin et al. [23], Hiles et al. [25], Willems and Briggs [27], Willems et al. [28]	$1.695\;\times \;\dot{V}$O_2_ − 1.701 × $\dot{V}$CO_2_	$4.210\;\times \;\dot{V}$CO_2_ − 2.962 × $\dot{V}$O_2_
Cook et al. [22]	$1.695\;\times \;\dot{V}$O_2_ − 1.701 × $\dot{V}$CO_2_	$4.344\;\times \;\dot{V}$CO_2_ − 3.061 × $\dot{V}$O_2_
Willems et al. [29]	$1.67\;\times \;\dot{V}$O_2_ − 1.67 × $\dot{V}$CO_2_	$4.55\;\times \;\dot{V}$CO_2_ − 3.21 × $\dot{V}$O_2_

## Data Availability

Data can be provided upon reasonable request from the corresponding author of this manuscript due to the need to seek permission from corresponding authors of the studies used in the secondary analysis.

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
