# Peer review of "Alterations of Exercise-Induced Carbohydrate and Fat Oxidation by Anthocyanin-Rich New Zealand Blackcurrant Are Associated with the Pre-Intervention Metabolic Function: A Secondary Analysis of Randomized Crossover Trials"

_nutrients, 2025, doi:10.3390/nu17060997_

Round 1
Reviewer 1 Report
Comments and Suggestions for Authors
The manuscript, as it stands, does not provide substantial novel findings but rather reports correlations with previously published data. I recommend considering publication as a commentary paper or integrating the findings into a systematic review.
Suggestions:
- The title and abstract lack clarity. It appears that readers must be familiar with prior studies to understand the secondary analysis. The abstract should be self-contained and clearly present the new findings.
- While the introduction provides a good explanation of metabolic energy sources and the respiratory exchange ratio, it lacks information on the composition of the New Zealand blackcurrant (NZBC) extract and the possible mechanisms of action. Including these details, particularly with reference to previous studies, would significantly strengthen the rationale for the study.
- The methods section is not well-structured. It should start with a 2.1 Study Design section, followed by 2.2 Statistical Analysis to improve clarity and logical flow.
- The discussion should be improved by including a more detailed physiological mechanistic explanation of NZBC’s effects. Additionally, gender-dependent metabolic responses should be addressed to provide a more comprehensive interpretation of the results.
- The conclusion is currently weak and should be strengthened to better reflect the study’s key findings and implications.
Reviewer 2 Report
Comments and Suggestions for Authors
The study is a secondary analysis of the relationship between control/placebo RER during exercise and RER during exercise after supplementation with blackcurrent extract. The comparison between males and females is a nice addition to the analysis.
Abstract conclusions: I think you need to be more specific with your conclusion statements. For example, you indicate that those with very low intrinsic RER show increase carbohydrate utilization and decreased fat utilization with the blackcurrent intervention. Do people with high intrinsic RER show the opposite effect?
Can you provide a hypothesis statement at the end of the introduction?
Line 111: Add a statement here that all participants signed informed consent forms.
Line 111: Please clarify how you had 141 observations from 69 males. Did some males participate in more than one of the studies?
Line 119: “exercise-induced induced substrate utilisation.” – delete the second “induced”.
Line 258: “Intuitively, for individuals that already have a high internal ability for fat oxidation, the mechanisms and potential changes for alteration of enhancing fat oxidation may be less sensitive to the effects by anthocyanins or anthocyanin-induced metabolites.” Did you consider looking at the correlation between fat oxidation while on placebo/control and while on the blackcurrent extract?
Endurance training usually lowers RER at a given workload. Is the blackcurrent supplement more effective for increasing fat oxidation during exercise in untrained than trained individuals?
Reviewer 3 Report
Comments and Suggestions for Authors
Evaluation of manuscript nutrients-3471206
This is an interesting study about the supplementation of anthocyanin-rich New Zealand blackcurrant and exercise in metabolic syndrome. I will present below my considerations point-by-point aiming to help the authors.
What is the likely physiological mechanism by which anthocyanin-rich New Zealand blackcurrant promotes ergogenic effect in exercise? This should be presented in the introduction.
Make clear what the original aspects of this study are.
Please, add the hypothesis after the aims of the study.
The discussion is the weakest point of the study:
1. Results should not be presented in the discussion.
2. The authors do not present the possible physiological mechanisms and effects of the supplement tested at any time.
3. How the dose-response effect occurs is not discussed. Why did previous studies select the different doses tested?
4. Limitations of the present study are not presented. Please review.
Round 2
Reviewer 1 Report
Comments and Suggestions for Authors
The study design requires further clarification to ensure transparency for readers. Specifically, the methodology should explicitly describe how patient data from different trials were merged, given that the interventions (e.g., timing, dosage, and exercise tasks) are not homogeneous. Addressing potential sources of variability and explaining how they were managed or adjusted for in the analysis would strengthen the study's validity
Author Response
Thanks again for taking the time to review the manuscript and our responses. We did not merge data but each datapoint from an individual in a study was taken to get the 141 male observations for example. We added the word “individual” to clarify, e.g. “The individual data of 47 females (47 observations) and 69 males (141 observations) were included in the secondary analysis.”
We also added “Note that the 141 observations were due to three male cohort studies with one having 7- and 14-day intake duration [23], one three different intensities [20] and one two doses [21] used for analysis.”
All the studies have differences in methodology and we did not attempt therefore to quantify absolute values of substrate utilization we agree that dose, exercise task etc would need to be considered.
The ‘simple’ aim of the manuscript was to have the take home message that divergent responses with respect to exercise-induced substrate utilization can occur by the intake of New Zealand blackcurrant extract due to the intrinsic/pre-intervention respiratory exchange ratio. This information will be useful for future research to examine the mechanisms of the effect of New Zealand blackcurrant extract on substrate utilization because as of now enhanced exercise-induced fat oxidation and lower carbohydrate oxidation were seen as the potential effect but our analysis shows that a divergent response is possible. We did not aim to do a multiple linear regression analysis.
Reviewer 3 Report
Comments and Suggestions for Authors
The revised version of nutrients-3471206 demonstrates the authors' applied efforts to address the reviewers' feedback and enhance the overall quality of the text. The authors have diligently incorporated nearly all the requested revisions, with the exception of one point regarding the study's limitations. Specifically, the authors did not explicitly acknowledge that conducting a secondary analysis inherently presents a limitation, as the data utilized were originally generated for other studies. If deemed appropriate by the editor, it would be valuable to request the authors to include this limitation in their discussion, as it provides important context for interpreting the study's findings.
Author Response
Thanks again for the time and engagement with reviewing the manuscript.
We have added as limitations.
“However, a few limitations need to be noted. First, none of the cohort and case studies specifically recruited individuals with a particular respiratory exchange ratio. Second, the data collection and analysis methods for obtaining the metabolic data was different between the studies. Third, none of the studies implemented a strict dietary control which may have affected the absorption, distribution, metabolism and excretion of anthocyanins and anthocyanin-induced metabolites. Fourth, age and training status were not considered. Notwithstanding that the precise mechanisms and factors that contribute to the alteration of exercise-induced substrate utilisation by intake of New Zealand blackcurrant are unknown, our secondary analysis indicates that the individual metabolic response is related to the intrinsic respiratory exchange ratio.”